# Assessment of Growth Differentiation Factor 15 Levels on Coronary Flow in Patients with STEMI Undergoing Primary PCI

**DOI:** 10.3390/diseases8020016

**Published:** 2020-05-25

**Authors:** Orhan Dogdu

**Affiliations:** Department of Cardiology, Medical Park Hospital, Elazig 23000, Turkey; orhandogdu@yahoo.com

**Keywords:** growth differentiation factor-15, ST elevation myocardial infarction, primary percutaneous coronary intervention

## Abstract

Growth Differentiation Factor-15 (GDF-15) is a strong predictor of decreased myocardial salvage and subsequent higher risk of death in patients with STEMI, but no information has been published regarding the association of GDF-15 levels with coronary blood flow in STEMI. We hypothesized that elevated GDF-15 levels would be associated with impaired flow and perfusion in the setting of STEMI treated with primary PCI. Eighty consecutive patients who were admitted with STEMI within 6 h from symptom onset were enrolled in the study. Patients were divided into two groups based upon the Thrombolysis in Myocardial Infarction (TIMI) flow grade. Group 1 was defined as TIMI Grade 0, 1 and 2 flows. Angiographic success was defined as TIMI 3 flow (group 2). GDF-15 and high sensitive CRP were measured. Major adverse cardiac events (MACE) were defined as stent thrombosis, nonfatal myocardial infarction and in-hospital mortality. There were 35 patients (mean age 64 ± 11.8 and 20% female) in group 1 and 45 patients (mean age 66.8 ± 11.5 and 29% female) in group 2. GDF-15 and hs-CRP levels were significantly higher in group 1 than in group 2 (1670 ± 831pg/mL vs. 733 ± 124 pg/mL, *p* < 0.001; and 19.8 ± 10.6 vs. 11.3 ± 4.9, *p* < 0.001). GDF-15 level ≥920 pg/mL measured on admission had a 94% sensitivity and 91% specificity in predicting no-reflow at ROC curve analysis. In-hospital MACE was also significantly higher in group 1 (28.6% vs. 2.2%, *p*: 0.001). Additionally, there was a significant correlation between hs-CRP and GDF-15 (r: 0.6030.56; *p* < 0.001). The GDF-15 level on admission is a strong and independent predictor of poor coronary blood flow following primary PCI and in hospital MACE among patients with STEMI. Except for predictive value, GDF-15 levels may be a useful biomarker for the stratification of risk in patients with STEMI, and may carry further therapeutic implications.

## 1. Introduction

Primary percutaneous coronary intervention (PCI) is a common treatment modality for patients with ST segment elevation myocardial infarction (STEMI) [1]. However, no-reflow is still a challenging problem in the management of patients with STEMI undergoing primary PCI. It is well known that angiographic no-reflow is strongly correlated with morbidity and mortality in acute STEMI. Rapid restoration of infarct-related arterial (IRA) flow is associated with improved ventricular performance and lower mortality and morbidity among patients with myocardial infarction [2,3]. The pathophysiology of no-reflow has not yet been completely explained, and its etiology appears to be multifactorial and very sophisticated [4].

Several biomarkers are associated with poor prognosis in STEMI. Increased Mean platelet Volume (MPV) levels have been associated with poor clinical outcome in survivors of myocardial infarction [5,6]. C-reactive protein (CRP) is an acute phase protein; several studies have shown that elevated CRP may have prognostic value in patients with acute coronary syndromes who are undergoing percutaneous coronary intervention [7,8,9].

Growth Differentiation Factor-15 (GDF-15) is a cytokine acting as a marker of oxidative stress; it plays a role in multiple diseases including cardiovascular disease, various cancers, renal failure, diabetes mellitus and inflammatory diseases [10]. Most studies on GDF-15 in cardiovascular disease have focused on coronary artery diseases, since this biomarker is strongly related to infarcted human heart [11,12]. In patients with non-ST-elevated acute coronary syndrome, a high GDF-15 level is a potent predictor of mortality and may be useful for decision making regarding an invasive treatment [13,14]. GDF-15 is also a strong predictor of decreased myocardial salvage and subsequent higher risk of death in patients with STEMI undergoing primary PCI [15]. In the present study, we hypothesized that elevated GDF-15 levels would be associated with impaired flow and perfusion in STEMI treated with primary PCI.

## 2. Materials and Methods

### 2.1. Study Population

Eighty consecutive patients (male 75% and mean age 65.5 ± 11.6) who were admitted with STEMI within 6 h from symptom onset were enrolled in the study. All of the patients were treated with primary PCI at our institution from September, 2015 to March, 2016. STEMI was defined as follows: typical chest pain >30 min duration with ST elevation >1 mm in at least two consecutive leads on the electrocardiogram. Patients were divided into two groups based upon the Thrombolysis In Myocardial Infarction (TIMI) flow grade score [16]. No-reflow was defined as TIMI grade 0, 1 and 2 flows (group 1) post-PCI. Angiographic success was defined as TIMI 3 flow (group 2) [17,18]. Exclusion criteria included treatment of STEMI in the previous 24 h with thrombolytic drugs, congenital heart disease, chronic renal failure, known malignancy, known inflammatory disease, infectious disease, hematological disease, autoimmune disease and end stage liver failure. This study complied with the Declaration of Helsinki; informed consent was obtained from all patients and the protocol was approved by Firat University Medical School Ethics Committee (14/07,2015).

### 2.2. Coronary Angiography and PCI Procedure

A conventional coronary angiography was performed with Philips Allura Xper FD10 equipment (Philips Medical Systems, Best, The Netherlands) in all patients after admission. All primary PCI procedures were performed with the standard femoral and radial approach with a 6- and 7-French guiding catheter. After administration of 5000 IU heparin and 600 mg clopidogrel loading dose conventional wire crossing, direct stenting was implanted whenever possible; in the remaining cases, balloon predilatation was carried out. The choice of stent (bare metal or drug-eluting) was at the operator’s discretion. In each patient treated with tirofiban, the drug was administrated after primary PCI procedure in a coronary care unit. The use of systemic bolus of tirofiban, followed by a 12 h continuous infusion, was at the operator’s discretion. In each patient treated with abciximab, the drug was administrated in catlab. The use of intracoronary (0.25 mg/kg IV bolus over at least 1 min) bolus of abciximab, followed by a continuous infusion (0.125 mcg/kg/min IV), was at the operator’s discretion. To achieve maximal dilatation, each coronary angiogram was preceded by intracoronary injection of 100 µg nitroglycerine. The TIMI grade was assessed by two independent interventional cardiologists.

### 2.3. Laboratory Analysis and Echocardiography

In all patients, antecubital venous blood samples for the laboratory analysis were drawn on admission in the emergency room. High sensitive CRP was measured using a BN2 model nephlometer (Dade-Behring). Common blood counting parameters were stored in citrate-based anticoagulated tubes and measured using a Sysmex K-1000 auto analyzer within 5 min of sampling. Serum creatinine, serum glucose, serum total cholesterol (TC), serum TG, and serum HDL-C were determined using colorimetric methods with the Roche assay (Roche cobas6000). Serum low density cholesterol (LDL-C) was indirectly calculated. We measured GDF-15 serum concentrations with a precommercial chemiluminescent microparticle immunoassay on a Hitachi cobas e411 analyzer (Roche Diagnostics, Mannheim, Germany).

A transthoracic echocardiography was performed for each patient immediately after primary PCI in the intensive cardiac care unit. All measurements were performed using a commercially available machine (Vivid E9^®^ GE Medical System, Horten, Norway) with a 3.5-MHz transducer.

### 2.4. Follow-Up and Major Adverse Cardiac Events

Major adverse cardiac events (MACE) were defined as in stent thrombosis, nonfatal myocardial infarction and in-hospital mortality during the in-hospital follow up period. In-stent thrombosis was defined as angiographically-documented total occlusion. Nonfatal myocardial infarction was defined as recurrent chest pain and/or development of new ECG changes accompanied by a new rise of ≥20% of cardiac biomarkers measured after the recurrent event. In-hospital mortality had to be verified death due to myocardial infarction, cardiac arrest or other cardiac causes.

### 2.5. Statistical Analysis

Continuous variables were tested for normal distribution by the Kolmogorov–Smirnov test. We report continuous data as mean and standard deviation or median. We compared continuous variables using a student t-test or Mann-Whitney U test between groups. Categorical variables were summarized as percentages and compared with the Chi-square test. Pearson correlation coefficients examined the degree of association between examined variables. *p* value <0.05 was considered as significant. The Receiver Operating Characteristics (ROC) curve was used to demonstrate the sensitivity and specificity of GDF-15, i.e., the optimal cut-off value for predicting poor coronary flow after primary PCI in patients with STEMI. *p* value <0.05 was considered as significant and confidence interval (CI) was 95%. All statistical analyses were performed with the SPSS version 21 (SPSS, Inc., Chicago, IL, USA).

## 3. Results

The baseline demographic, biochemical characteristics, history of medicine use and angiographic properties of patients in both groups are shown in Table 1. There were 35 patients (mean age 64 ± 11.8 and 20% female) in group 1 and 163 patients (mean age 66.8 ± 11.5 and 29% female) in group 2. With respect to coronary risk factors, there was a significant difference in the presence of diabetes mellitus (DM) (*p*: 0.02), but there was no significant difference in hypertension, prior coronary artery disease, hypercholesterolemia and smoking status (*p*: 0.68, *p*: 0.39, *p*: 0.08, *p*: 0.92 respectively).

With respect to baseline laboratory status, the serum glucose concentration on admission was significantly higher in group 1 (*p*: 0.008), while there was no significant difference in serum lipid profile, blood urea nitrogen, creatinine, hemoglobin (Hg), platelet and white blood cell count (WBC) between groups. However, hs-CRP levels were significantly higher in group 1 than in group 2 (19.8 ± 10.6 vs. 11.3 ± 4.9, *p* < 0.001). Additionally, left ventricular ejection fraction (LVEF) and pain to balloon time were not significantly different between groups (*p*: 0.14 and *p*: 0.36, respectively) (Table 1).

In hemodynamic parameters on admission of patients, there was no significant differences in systolic and diastolic blood pressures and heart rate between two groups (*p*: 0.46, *p*: 0.43, and *p*: 0.53, respectively) (Table 1).

A greater proportion of patients with multivessel disease were in group 1 (*p*: 0.002). There was no significant difference in involvement of circumflex, right coronary artery, left anterior descending artery or saphenous graft as infarct-related artery (IRA) between groups (*p*: 0.07, *p*: 0.43, *p*: 0.09 and *p*: 0.45, respectively). In the PCI procedure, stent implantation percentage and the used stent types were similar between groups (*p* > 0.05) (Table 1).

In the previous medication histories of patients, there was no significant difference in angiotensin converting enzyme inhibitors/angiotensin receptor blockers, b-blocker, statin, aspirine and diuretic drugs usage between groups (*p*: 0.54, *p*: 0.15, *p*: 0.89, *p*: 0.51 and *p*: 0.70, respectively) (Table 1).

GDF-15 levels were significantly higher in group 1 than in group 2 (1670 ± 831pg/mL vs. 733 ± 124 pg/mL, *p* < 0.001) (Table 1).

In-stent thrombosis, nonfatal MI and in-hospital mortality were significantly higher in group 1 (*p*: 0.04, *p*: 0.02, and *p*: 0.02, respectively). Overall, in-hospital MACE was also significantly higher in group 1 (28.6% vs. 2.2%, *p*: 0.001) (Table 1).

The ROC curves of GDF-15 for predicting no-reflow are shown in Figure 1A. GDF-15 level ≥920 pg/mL measured on admission had a 94% sensitivity and 91% specificity in predicting no-reflow.

There was a significant correlation between hs-CRP and GDF-15 (r: 0.6030.56; *p* < 0.001) (Figure 2).

Additionaly, the ROC curves of hs-CRP for predicting no-reflow are shown in Figure 1B. Hs-CRP level ≥13.2 mg/L measured on admission had a 65% sensitivity and 59% specificity in predicting no-reflow. Thus, GDF-15 is a superior marker to hs-CRP for predicting TIMI < 3.

When we divided the study population into two groups according to the 920 pg/mL GDF-15 level cut-off value used in the ROC analysis, multivessel disease and no-reflow phenomena were significantly higher in the increased uric acid group (*p*: 0.012, *p*: 0.005, respectively). Hs-CRP levels were significantly higher in the increased GDF-15 group than in the other group (13.6 ± 7.3 mg/L vs. 39.2 ± 14.5 mg/L, *p*: 0.002). In-hospital MACE was significantly higher in the increased GDF-15 group (*p* < 0.001). Age and sex were not significantly different between groups (*p*: 0.22 and *p*: 0.30 for male sex, respectively). With respect to coronary risk factors, hypercholesterolemia and diabetes mellitus were significantly higher in the increased GDF-15 group (*p*: 0.024, *p*: 0.039, respectively) (Table 2).

Some variables that can influence coronary flow were significantly different between groups. Thus, the effects of multiple variables on the coronary flow were analyzed with univariate and multivariate logistic regression analyses. The variables for which the unadjusted *p* value was <0.10 in the univariate analysis were identified as potential risk markers for angiographic no-reflow, and were included in the full model. At multivariate analysis, GDF-15 level (odds ratio (OR): 1.018, 95% confidence interval (CI) 1.007–1.029; *p* = 0.018) was an important risk factor of coronary flow in patients with STEMI undergoing primary PCI (Table 3).

## 4. Discussion

This study has demonstrated three main results: First, there is a significant relationship between serum GDF-15 levels and postprimary PCI myocardial perfusion grade; Second, serum GDF-15 is a specific and sensitive predictor of poor coronary blood flow after primary PCI in STEMI; And third, hs-CRP and GDF-15 are correlated with each other (Figure 3).

In previous studies, increased GDF-15 levels were associated with MI, stroke and cardiovascular death, as well as with reduced LVEF, coronary artery disease and HF in the elderly [19,20,21]. Additionally, GDF-15 was reported to be independently related to adverse events in STEMI and non-ST-elevation acute coronary syndrome [13,22,23]. In our study, we demonstrated that GDF-15 level is a predictor of poor coronary blood flow after primary PCI in STEMI and in hospital MACE. The possible effect of GDF-15 may be explained partly by its association with inflammatory processes. In arteriosclerotic carotid arteries, the immunoreactivity of GDF-15 was shown to combine with the immunoreactivities of apoptosis markers in the active macrophage [24,25]. Conversely, GDF-15 appeared to have anti-inflammatory and antiapoptotic effects in the heart tissue [26,27]. Myocardial GDF-15 levels increased quickly after ischemic injury and seemed to protect myocardium from reperfusion injury in an experimental model [28]. GDF-15-insufficient mice were found to have enhanced recruitment of polymorphonuclear leukocytes and increased incidence of cardiac rupture after MI. Furthermore, infusion of recombinant GDF-15 inhibited polymorphonuclear leukocyte recruitment in GDF-15-insufficient mice [29]. It is likely that increased GDF-15 levels are induced by upstream proinflammatory cytokines, but represent an endogenous protective effort trying to limit cardiovascular damage [30,31]. Future studies on the pathophysiology of GDF-15 in MI should be done before using GDF-15 as a therapeutic target. Eitel et al. demonstrated that GDF-15 is a strong predictor of mortality in patients with STEMI reperfused by primary PCI that provides prognostic information about established clinical outcome cardiac MR parameters. This study provides further insights into the pathophysiological mechanisms of GDF-15, supporting the hypothesis that GDF-15 integrates information from different disease pathways in acute STEMI and provides unique additional prognostic information [15,32]. The present study demonstrated that GDF-15 ≥920 pg/mL predicts no-reflow with 94% sensitivity and 91% specificity. Additionally, elevated GDF-15 levels were also a strong and independent predictor of no-reflow and in-hospital MACE in patients with STEMI undergoing primary PCI.

The pathophysiology of no-reflow has not been completely explained and its etiology appears to be multifactorial and very sophisticated. These factors include ischemic endothelial damage, platelet plugging and microvascular leukocytes, reactive oxygen species and complex interactions between leukocytes and platelets induced by the inflammatory process [18]. There is also a relationship between the recovery of left ventricular function, morbidity and mortality after an acute myocardial infarction and the no-reflow phenomenon [4].

With understanding of the role of inflammation in the atherosclerotic process, studies have focused on hs-CRP as an importantly marker of risk. Hs-CRP is a marker of inflammation with a half-life of 19 h; it is released 6 h after a coronary event, on average. Elevation of hs-CRP has been shown in acute coronary syndromes, and it has been demonstrated to be associated with cardiac events [5]. Several studies have shown a significant correlation between the vascular occlusion grade and baseline hs-CRP levels. Inflammation has also been implicated in the development of the no-reflow phenomenon [8,9]. Tomoda et al. demonstrated that CRP levels within 6 h of the onset of acute myocardial infarction can predict both the vulnerability of culprit coronary lesions and the likelihood of adverse coronary events after primary PCI [8]. GDF-15 levels are independently related to cardiovascular risk factors (diabetes, smoking, low HDL cholesterol) and biochemical risk markers (high-sensitivity C-reactive protein, NT-pro BNP) in elderly individuals and patients with coronary heart disease [20]. Additionaly, the magnitude of adjusted risk is similar or greater than that associated with CRP in primary prevention trials [33]. These findings suggest that GDF-15 may be a marker of pathobiological mechanisms which is distinct from those identified by hs CRP. Due to their anti-inflammatory properties, it has been proposed that statins may have the potential to influence GDF-15 and its relationship to risk, as has been observed with hsCRP [30]. Continued research should focus on improving understanding of the pathobiological features of GDF-15 and its role in myocardial tissue injury. Such research may reveal new treatment targets related to GDF-15 in patients stabilized after a STEMI and help to better define the clinical role of this emerging biomarker.

## 5. Limitations

The major limitation of the present study is that it was carried out within a single center, as shown by the very small number of patients; however, our results are relevant to inspire for further studies with larger sample sizes. Additionaly, our population contained heterogeneous unselected STEMI patients. Therefore, these data may not be generalized to the total STEMI population. Future studies are needed to validate our findings in STEMI and determine their generalizability to other populations and ethnicities. Thus, the results of this study require larger studies with more biomarkers.

## 6. Conclusions

GDF-15 level on admission is a strong and independent predictor of poor coronary blood flow following primary PCI. Increased GDF-15 levels are also a strong and independent predictor of no-reflow and in-hospital MACE. Apart from predictive value, GDF-15 levels may help to better define the clinical role of this emerging biomarker.

## Figures and Tables

**Figure 1 diseases-08-00016-f001:**
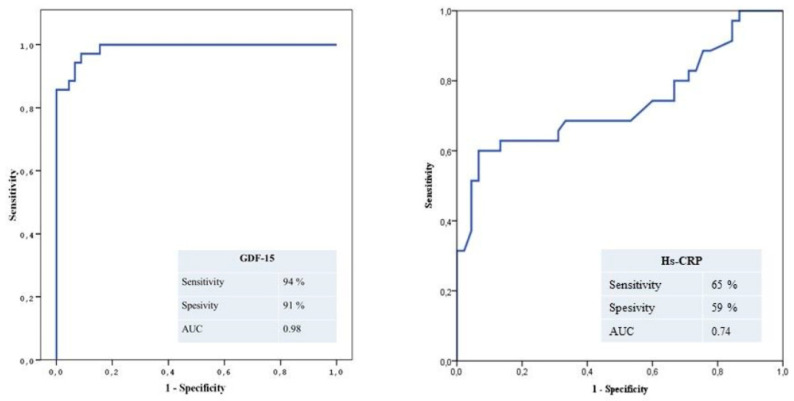
(**A**) The receiver-operating characteristic (ROC) curve of GDF-15 for predicting angiographic no-reflow. (**B**) ROC curve of Hs-CRP for predicting angiographic no-reflow.

**Figure 2 diseases-08-00016-f002:**
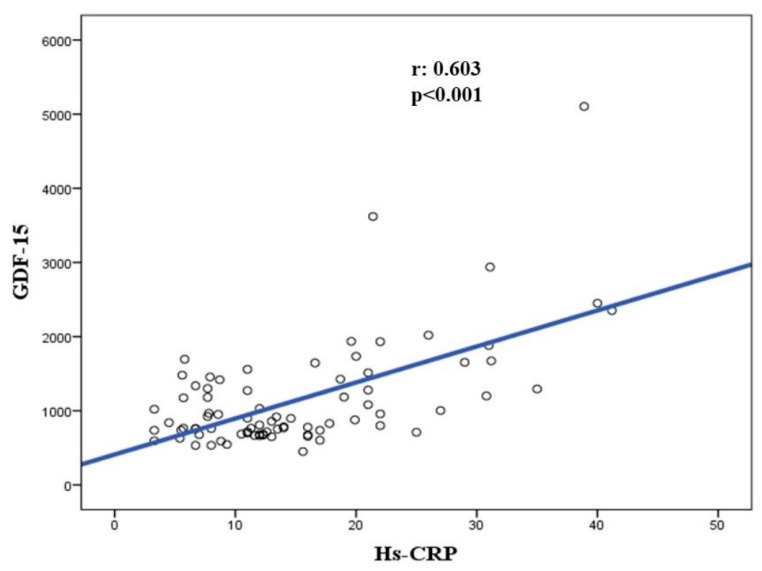
Correlation between hs-CRP and GDF-15 levels.

**Figure 3 diseases-08-00016-f003:**
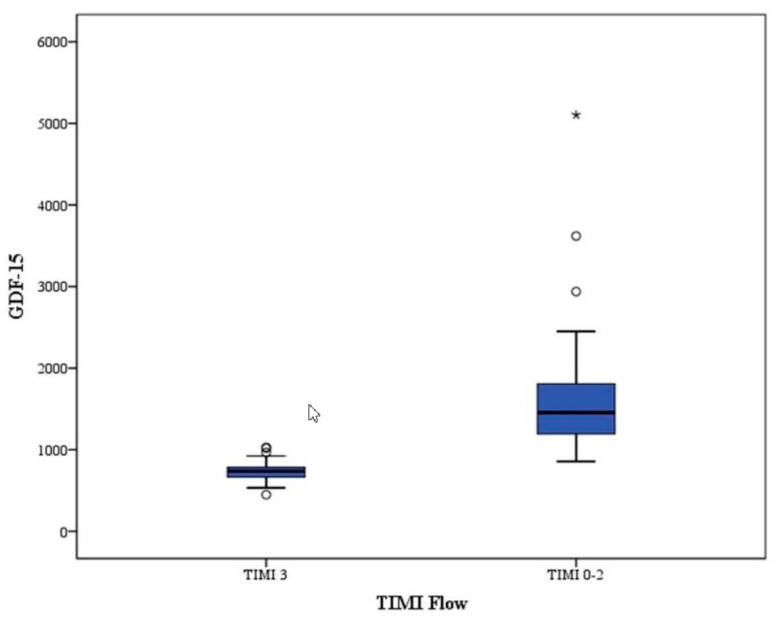
Box-plot between GDF-15 and TIMI Flow.

**Table 1 diseases-08-00016-t001:** Baseline characteristics of patients in groups.

Variable	Group 1 (TIMI 0-2) (n:35)	Group 2 (TIMI 3) (n:45)	*p* Value
Age, years	64 ± 11.8	66.8 ± 11.5	0.29
Gender, Female/Male	7/28	13/32	0.36
Body mass index, kg/m^2^	26.6 ± 2.5	25.6 ± 2.7	0.08
Previous CAD, n (%)	10 (28.6)	17 (37.8)	0.39
Smoking, n (%)	12 (34.3)	15 (33.3)	0.92
Hypertension, n (%)	14 (40)	16 (35.6)	0.68
Hypercholesterolemia, n (%)	8 (22.9)	4 (8.9)	0.08
Diabetes mellitus, n (%)	17 (48.6)	11 (24.4)	0.02
Blood Pressure on admission (mmHg)			
Systolic	126.5 ± 21.1	130.1 ± 21.5	0.46
Diastolic	76.7 ± 16.6	79.6 ± 15.9	0.43
Heart rate, beats/min	88.2 ± 17.8	85.9 ± 14.9	0.53
Biochemical parameters			
Total cholesterol, mg/dl	174.6 ± 36.5	174.3 ± 32.6	0.97
HDL-cholesterol, mg/dl	36.6 ± 6.5	39.5 ± 11.1	0.18
LDL-cholesterol, mg/dl	109.1 ± 32.8	112.4 ± 31.4	0.67
Serum triglycerides, mg/dl	147.3 ± 84.9	140.9 ± 76.5	0.79
Serum glucose, mg/dL	163.8 ± 66.7	123.7 ± 45.4	0.008
Blood urea nitrogen, mg/dL	44.5 ± 13.7	44.7 ± 23.9	0.95
Creatinine, mg/dL	0.92 ± 0.22	0.86 ± 0.43	0.44
Hs-CRP(mg/L)	19.8 ± 10.6	11.3 ± 4.9	<0.001
LVEF on admission	45.1 ± 7.5	47.7 ± 6.2	0.14
Hemoglobine (g/dl)	14 ± 1.8	13.3 ± 3	0.20
White blood cell count, × 10^9^/L	10.7 ± 2.9	10.9 ± 3.9	0.82
Platelet count, × 10^9^/L	244 ± 84	232 ± 79	0.53
Previous medications, n (%)			
Aspirin	10 (28.6)	16 (35.6)	0.51
Beta-blockers	6 (17.1)	14 (31.1)	0.15
ACE-inhibitors/ARB	8 (22.9)	13 (28.9)	0.54
Statins	9 (25.7)	11 (24.4)	0.89
Ca-antagonists	6 (17.1)	4 (8.9)	0.27
Diuretics	4 (11.4)	4 (8.9)	0.70
Glycoprotein IIb/IIIa antagonist	10 (28.6)	5 (11.1)	0.04
Pain to balloon time (h)	4.2 ± 0.8	4.0 ± 1.0	0.36
Hospitalization (day)	7.1 ± 1.3	6.1 ± 1.2	0.001
Infarct related artery, n (%)			
RCA	21 (60)	23 (51.1)	0.43
LAD	13 (37.1)	9 (20.1)	0.09
Cx	22 (62.9)	19 (42.2)	0.07
Saphenous graft or LIMA	3 (8.6)	2 (4.4)	0.45
Coronary artery involvement			
Single-vessel disease	17 (48.6)	37 (82.2)	0.002
Multivessel disease	18 (51.4)	8 (17.8)	0.002
Primery PCI			
Stent implantation, n (%)	34 (97.1)	43 (95.6)	0.71
BMS, n (%)	8 (22.9)	16 (38.1)	0.15
DES, n (%)	26 (74.3)	28 (62.2)	0.25
Stent lenght (mm)	20.9 ± 7.4	19.8 ± 8.3	0.65
Stent diameter (mm)	2.95 ± 0.5	2.91 ± 0.4	0.55
GDF-15, pg/mL	1670 ± 831	733 ± 124	<0.001
In-hospital MACE, n (%)	10 (28.6)	1 (2.2)	0.001
In stent thrombosis	5 (14.3)	1 (2.2)	0.04
Nonfatal MI	6 (17.1)	1 (2.2)	0.02
In-hospital mortality	4 (11.4)	0	0.02

Data expressed mean ± SD and percentage (%) for categorical variables. CAD: coronary arterial disease, CRP: C-reactive protein, LVEF: left ventricular ejection fraction, LDL: low density lypoprotein, HDL: high density lypoprotein, WBC: white blood cell, BMI: Body mass index, ACEi: angiotensin converting enzyme inhibitors, LAD: left anterior descending, CX: circumflex artery, RCA: right coronary artery, LIMA: left internal mammarian artery, PCI: percutaneous coronary intervention, BMS: bare metal stent, DES: drug eluting stent, MACE: major advanced cardiovasculary events, MI: myocardial infarction.

**Table 2 diseases-08-00016-t002:** Baseline risk factors and in-hospital MACE stratified by GDF-15 levels.

Variable	GDF-15 < 920 (n:44)	GDF-15 ≥ 920 (n:36)	*p* Value
Age, years	67 ± 11.3	63.8 ± 11.9	0.22
Gender, Female/Male	13/31	7/29	0.30
Coronary risk factors			
Previous CAD, n (%)	16 (36.4)	11 (30.6)	0.58
Smoking, n (%)	14 (31.8)	13 (36.1)	0.68
Hypertension, n (%)	16 (36.4)	14 (38.9)	0.81
Hypercholesterolemia, n (%)	3 (6.8)	9 (25)	0.024
Diabetes mellitus, n (%)	11 (25)	17 (47.2)	0.039
Severity of CAD			
Single-vessel disease	35 (79.5)	19 (52.8)	0.012
Multivessel disease	9 (20.5)	17 (47.2)	0.012
No-reflow, n (%)	0	6 (16.7)	0.005
In-hospital MACE, n (%)	0	11 (30.6)	<0.001
In stent thrombosis	0	6 (16.7)	0.005
Nonfatal MI	0	7 (19.4)	0.002
In-hospital mortality	0	4 (11.1)	0.024

Data expressed mean ± SD. CAD: coronary arterial disease, MACE: major advanced cardiovasculary events, MI: myocardial infarction.

**Table 3 diseases-08-00016-t003:** Effects of variables on angiographic no-reflow in univariate and multivariate logistic regression analyses.

Variables	Unadjusted OR	95% CI	*p* Value	Adjusted OR *	95% CI	*p* Value
Age	0.979	0.942–1.018	0.292			
Diabetes mellitus	2.919	1.130–7.545	0.027	1.488	0.062–35.950	0.807
Heart Rate	1.009	0.982–1.037	0.519			
BMI	1.165	0.975–1.391	0.093			
Hemoglobine	1.120	0.928–1.352	0.238			
LAD lesion	2.364	0.868–6.437	0.092			
Hs-CRP	1.148	1.067–1.236	<0.001	1.309	0.896–1.913	0.164
Multivessel disease	4.897	1.781–13.467	0.002	18.85	0.720–493.4	0.078
GDF-15	1.018	1.007–1.029	0.001	1.021	1.004–1.038	0.018
DES implantation	1.754	0.666–4.619	0.255			
Glycoprotein IIb/IIIa antagonist	3.200	0.979–10.457	0.054			

* Adjusted for, age, diabetes mellitus, heart rate, BMI, hemoglobine, LAD lesion, CRP, multivessel disease, GDF-15, DES implantation and glycoprotein IIb/IIIa antagonist. OR: Odds ratio, CI: Confidence interval.

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
