# Peer review of "Assessment of Growth Differentiation Factor 15 Levels on Coronary Flow in Patients with STEMI Undergoing Primary PCI"

_diseases, 2020, doi:10.3390/diseases8020016_

Round 1

Reviewer 1 Report

Dogdu sought to investigate the association of GDF-15 with coronary blood flow in STEMI patients. The author enrolled 80 patients and they were divided in 2 subgroups (TIMI flow 0-2 and TIMI flow 3).

Table 1: I think „primary PCI“ is meant here?

The author could try to performed a combined analysis of GDF-15 and CRP, e.g. both GDF-15 and CRP above cut-off compared to GDF-15 and CRP below cut-off.

The author could show a comparison of GDF-15 in TIMI flow 0-2 and TIMI flow 3 using e.g. a box-plot.

Please address why patients were treated with clopidogrel instead of ticagrelor or prasugrel.

Please describe how GDF-15 was measured (ELISA?), please describe the method.

Author Response

Response to Reviewer 1:

  1. You requested that ‘Table 1: I think „primary PCI“ is meant here?.

We have made a mistake as you noticed. We take refuge in your understandings and we have made following changes;

We changed the Table 1 as follow;

"Primary PCI "

  1. You requested that ‘The author could try to performed a combined analysis of GDF-15 and CRP, e.g. both GDF-15 and CRP above cut-off compared to GDF-15 and CRP below cut-off.

We added the following sentences into the result section and table 2 as follow;

“Hs-CRP levels were significantly higher increased GDF-15 group than in the other group (13.6 ± 7.3 mg/L vs 39.2 ± 14.5 mg/L, p:0.002).”

  1. You requested that ‘The author could show a comparison of GDF-15 in TIMI flow 0-2 and TIMI flow 3 using e.g. a box-plot.

We have been added into manuscript as Figure 3.

  1. You requested that ‘Please address why patients were treated with clopidogrel instead of ticagrelor or prasugrel.

We are pleased with this constructive criticism. In Turkey, clopidogrel is cheaper than ticagrelor or prasugrel. Clopidogrel is more advantageous in terms of price and reliability ratio.

Materials and methods

  1. You requested that ‘Please describe how GDF-15 was measured (ELISA?), please describe the method.

We added the following sentence into laboratory analysis and echocardiography section of the Materials and Methods;

“We measured GDF-15 plasma concentrations with a pre-commercial chemiluminescent microparticle immunoassay on a Hitachi cobas e411 analyzer (Roche Diagnostics, Mannheim, Germany).”

Reviewer 2 Report

Dear Author,

The study fits well to the current trend on GDF-15 role for risk stratification in a variety of populations. Therefore, assessment of its concentrations in a subset of STEMI patients undergoing PCI represents a correct approach.

Major comments:

  1. Page 2, line 85. The issues regarding laboratory measurements are lacking or minimized. There is no information how GDF-15 was measured, which is the main biomarker measured in this study. Which method? Fresh or frozen samples were used? Serum or plasma? Moreover, other methods are described poorly, for example other biochemical parameters "were determined by standard methods". What is the standard method? Could you clarify?
  2. Page 3, line 122, Table 1. Previously you said that lipid profile parameters you measured in serum, but in the Table 1 you mention Plasma triglycerides? 
  3. Serum glucose concentration rather than level, it should be corrected accordingly.
  4. Page 6, line 162, wrong units for CRP
  5. Page 7, line 187-190 Discussion. firstly, secondly, etc. The correlation of CRP with GDF-15 is obvious and should be elimanted as a major result, but discussed in the text.
  6. Page 8, line 233-238 Limitations. The population is rather heterogenous with unselected STEMI patients undergoing PCI and results can not be transferred to the community. Morovere, results of the study are strongly affected by diabates as half of the group 1 and 1/4 of group 2 has DM2 ( also: Table 3, the highest OR is for the presence of DM). This should be mentioned in the Limitations and discussed in the Discussion. 
  7. Page 8, line 240-245. Conclusion should be shortened and improved. 

Minor comments:

  1. English corrections needed.
  2. Figure 1 correct for "Specificity" in part A and B.

Author Response

Response to Reviewer 2:

Major comments:

  1. You requested that ‘Page 2, line 85. The issues regarding laboratory measurements are lacking or minimized. There is no information how GDF-15 was measured, which is the main biomarker measured in this study. Which method? Fresh or frozen samples were used? Serum or plasma? Moreover, other methods are described poorly, for example other biochemical parameters "were determined by standard methods". What is the standard method? Could you clarify?

We added the following sentence into laboratory analysis and echocardiography section of the Materials and Methods;

“We measured GDF-15 plasma concentrations with a pre-commercial chemiluminescent microparticle immunoassay on a Hitachi cobas e411 analyzer (Roche Diagnostics, Mannheim, Germany).”

We changed the following sentence into laboratory analysis and echocardiography section of the Materials and Methods;

“Serum creatinine, serum glucose, serum total cholesterol (TC), serum TG, and serum HDL-C were determined using colorimetric methods with the Roche assay (Roche cobas6000). Serum low density cholesterol (LDL-C) was indirectly calculated.”

  1. You requested that ‘Page 3, line 122, Table 1. Previously you said that lipid profile parameters you measured in serum, but in the Table 1 you mention Plasma triglycerides?’

We have made a mistake as you noticed. We take refuge in your understandings and we have made following changes;

We changed the Table 1 as follow;

" Serum triglycerides "

  1. You requested that ‘Serum glucose concentration rather than level, it should be corrected accordingly.

We corrected this expression as serum glucose concentration in Manuscript.

  1. You requested that ‘Page 6, line 162, wrong units for CRP’

We corrected units for CRP as mg/L in Manuscript.

  1. You requested that ‘Page 7, line 187-190 Discussion. firstly, secondly, etc. The correlation of CRP with GDF-15 is obvious and should be elimanted as a major result, but discussed in the text’.

We added the following sentences into the discussion and references section as follow;

“GDF-15 levels are independently related to cardiovascular risk factors (diabetes, smoking, low HDL cholesterol) and biochemical risk markers (high-sensitivity C-reactive protein, NT-pro BNP) in elderly individuals and patients with coronary heart disease [20]. Additionaly, the magnitude of adjusted risk is similar or greater to that associated with CRP in primary prevention trials [33]. These findings suggest that GDF-15 may be a marker of pathobiological mechanisms distinct from those identified by hs CRP. Due to their anti-inflammatory properties, statins have been raised as having the potential to influence GDF-15 and its relationship to risk, as has been observed with hsCRP [30]. Continued research should focus on an improved understanding of the pathobiological features of GDF-15 and role in myocardial tissue injury. Such research may reveal new treatment targets related to GDF-15 in patients stabilized after an STEMI and help to better define the clinical role of this emerging biomarker.”

  1. Ridker PM, MacFadyen J, Libby P, Glynn RJ. Relation of baseline high-sensitivity C-reactive protein level to cardiovascular outcomes with rosuvastatin in the justification for use of statins in prevention: an intervention trial evaluating rosuvastatin (JUPITER). Am J Cardiol. 2010;106: 204–209.
  2. You requested that ‘Page 8, line 233-238 Limitations. The population is rather heterogenous with unselected STEMI patients undergoing PCI and results can not be transferred to the community. Morovere, results of the study are strongly affected by diabates as half of the group 1 and 1/4 of group 2 has DM2 ( also: Table 3, the highest OR is for the presence of DM). This should be mentioned in the Limitations and discussed in the Discussion.

We have been rearranged following sentences into The Limitations section;

“Additionaly, our population contain heterogeneous unselected STEMI patients. Therefore, these data may not be generalized to the total STEMI population. Future studies are needed to validate our findings in STEMI and determine their generalizability to other populations and ethnicities. Thus, the results of this study require larger studies with more biomarkers.”

  1. You requested that ‘Page 8, line 240-245. Conclusion should be shortened and improved.

We have been rearranged following sentences into The Conclusion section;

“GDF-15 levels on admission are strong and independent predictor of poor coronary blood flow following primary PCI. Increased GDF-15 levels are also a strong and independent predictor of no-reflow and in-hospital MACE. Apart from predictive value, GDF-15 levels may help to better define the clinical role of this emerging biomarker.”

Minor comments:

  1. You mentioned that ‘English corrections needed.’

We corrected English errors in the text as you requested. We have made a mistake as you noticed. We take refuge in your understandings and we have made specified changes.

  1. You requested that ‘Figure 1 correct for "Specificity" in part A and B.’

            We corrected Figure 1 as you requested.

Round 2

Reviewer 1 Report

The authors have responded to all queries.

Reviewer 2 Report

I have no further comments.